# Revisiting Host-Pathogen Interactions in Cystic Fibrosis Lungs in the Era of CFTR Modulators

**DOI:** 10.3390/ijms24055010

**Published:** 2023-03-05

**Authors:** Carla M. P. Ribeiro, Matthew G. Higgs, Marianne S. Muhlebach, Matthew C. Wolfgang, Monica Borgatti, Ilaria Lampronti, Giulio Cabrini

**Affiliations:** 1Marsico Lung Institute/Cystic Fibrosis Research Center, The University of North Carolina at Chapel Hill, Chapel Hill, NC 27599, USA; 2Department of Medicine, The University of North Carolina at Chapel Hill, Chapel Hill, NC 27599, USA; 3Department of Cell Biology and Physiology, The University of North Carolina at Chapel Hill, Chapel Hill, NC 27599, USA; 4Department of Microbiology and Immunology, The University of North Carolina at Chapel Hill, Chapel Hill, NC 27599, USA; 5Department of Pediatrics, The University of North Carolina at Chapel Hill, Chapel Hill, NC 27599, USA; 6Department of Life Sciences and Biotechnology, University of Ferrara, 44121 Ferrara, Italy; 7Innthera4CF, Center on Innovative Therapies for Cystic Fibrosis, University of Ferrara, 44121 Ferrara, Italy

**Keywords:** cystic fibrosis, airway epithelia, airway infection, airway inflammation, *Pseudomonas aeruginosa*, *Staphylococcus aureus*, CFTR modulators

## Abstract

Cystic fibrosis transmembrane conductance regulator (CFTR) modulators, a new series of therapeutics that correct and potentiate some classes of mutations of the CFTR, have provided a great therapeutic advantage to people with cystic fibrosis (pwCF). The main hindrances of the present CFTR modulators are related to their limitations in reducing chronic lung bacterial infection and inflammation, the main causes of pulmonary tissue damage and progressive respiratory insufficiency, particularly in adults with CF. Here, the most debated issues of the pulmonary bacterial infection and inflammatory processes in pwCF are revisited. Special attention is given to the mechanisms favoring the bacterial infection of pwCF, the progressive adaptation of *Pseudomonas aeruginosa* and its interplay with *Staphylococcus aureus*, the cross-talk among bacteria, the bronchial epithelial cells and the phagocytes of the host immune defenses. The most recent findings of the effect of CFTR modulators on bacterial infection and the inflammatory process are also presented to provide critical hints towards the identification of relevant therapeutic targets to overcome the respiratory pathology of pwCF.

## 1. Introduction

Cystic fibrosis (CF) is an autosomal recessive genetic disease caused by mutations of the CF *Transmembrane conductance Regulator* (*CFTR*) gene, which encodes an integral membrane ion transporter [1,2,3]. A large series of *CFTR* gene variants has been identified, and several hundred mutations have been confirmed to cause CF disease [4]. *CFTR* gene mutations alter the CFTR protein biology in different ways, which have been summarized into six, or sometimes seven, major classes. These include when CFTR protein expression is completely absent or massively reduced, when CFTR intracellular processing leads to degradation instead of maturation and correct localization, when CFTR ion conductance is reduced or completely inactivated, and when CFTR stability and the half-life on the plasma membrane are reduced [5]. Mutated CFTR protein affects the function of several organs, with a phenotype ranging between mild and severe. The exocrine pancreas, liver, intestinal tract, salivary glands, male reproductive organ, sweat glands and lungs are the sites of classical CF disease manifestations. The involvement of the respiratory tract, where recurrent infections begin in childhood and follow a succession of different bacteria, first by *Staphylococcus aureus*, followed by chronic infection with the opportunistic bacterium *Pseudomonas aeruginosa*, remains the primary cause of morbidity and the reduction in life expectancy [6]. Since the initial description of using anti-staphylococcal therapy for CF [7], tremendous advancements have been accomplished regarding the molecular pathogenesis of CFTR and the resulting organ pathology.

The achievements of CF basic science have allowed for the development of new therapies that correct and potentiate, at least partially, some classes of CFTR mutations. These therapies, termed “CFTR modulators” are being utilized in the clinic for pwCF and represent a real new era for the management of this disease [8]. However, the main hindrances of the present CFTR modulators are related to their limited efficacy in reducing the chronic lung bacterial infection burden and inflammatory response, which are the main causes of the pulmonary tissue damage and progressive respiratory insufficiency that lead to heavy daily therapies, limitations in the quality of life and premature death of pwCF. Lessons from CF research suggest not only to proceed rapidly from the lab bench to the bedside, but also to go back to bench to learn more when clinical applications show their limits. In this review, we discuss the cross-talk between the bacterial infection and lung host response in CF, with a focus on the molecular targets for innovative therapeutic approaches to combat CF lung infection and inflammation. 

## 2. Why Does CF Disease Favor Bacterial Infection?

### 2.1. The Basic Ion Defect in CF Influences the Composition of the Airway Surface Liquid 

The CFTR protein is an integral membrane transporter of chloride and bicarbonate ions, regulated by ATP binding and dependent on protein kinase A and C phosphorylation [9,10]. In the airway mucosa, the CFTR protein is localized on the apical membrane of different types of epithelial cells lining the lumen of airways and submucosal glands [11]. In addition to the known CFTR expression in bronchial epithelia, recent studies have revealed that CFTR is highly expressed in the superficial epithelia of the small airways [12], where it mediates the fluid secretory responses. These studies indicated that CFTR expression and CFTR-dependent ion transport processes are present in the secretory club cells in the large and small (<2 mm in diameter) airways, the latter being the earliest and most affected site in CF lung disease [13,14]. Together with secreting chloride and bicarbonate, CFTR inhibits the activation of the Epithelial Sodium Channel (ENaC) [15] and regulates other chloride channels [16]. Thus, mutations in CFTR protein lead to reduced chloride and bicarbonate secretion and excessive sodium reabsorption, promoting alterations of the transepithelial fluid transport and leading to airway surface liquid (ASL) dehydration [17].

### 2.2. The Mucociliary Clearance in Healthy and CF Lungs 

In normal airways, the ASL consists of a hydro-gel layer with gel-forming mucins, with a 2% solids-to-water content ratio (for review see [18]). Dust and microorganisms are trapped in the mesh of filamentous mucins and are rapidly transported from the distal airways to the trachea by ciliary beating. This process constitutes the very basic innate defense mechanism in the lungs, known as mucociliary clearance (MCC). This escalator complex is constituted by two functional hydro-gel layers: cilia are surrounded by a prevalently watery medium containing mucins MUC1, MUC4 and MUC16; the tips of the cilia are topped with a second mucus layer composed of water, globular proteins and predominantly mucin MUC5B. This steady-state condition is modified by pulmonary stress (e.g., viral or bacterial overload), where MUC5B is superimposed by mucin MUC5AC secreted by the goblet cells, which increases the solid component of the hydro-gel. In the absence of chronic broncho-obstructive diseases, the increase in mucin secretion under infective stress is alerted by cilial mechanosensing, which induces ion and water secretion mainly through the ATP/purinergic receptors interactions and transmembrane signaling that activates the calcium-dependent chloride channels and re-equilibrates the mucin/water ratio in the ASL. In CF, defective ion transport does not allow for the compensatory re-hydration of the excessive mucin secretion; as a result, the solid fraction of the mucus hydrogel is increased, and the upper mucus layer compresses the cilia, slowing the beating frequency. In the case of severe ion transport defects, hyperconcentrated mucus flattens the cilia, which promotes the retention of microorganisms, rather than their clearance. Dehydrated airway mucus with adherent microorganisms constitutes the mucus plaques, which plug small bronchioles and activate an infective/inflammatory “vicious cycle”; this process involves the resident macrophages and epithelial cells, leading to the massive recruitment of neutrophils, which fill the CF bronchiolar lumen from the early months of life of CF infants [19]. Importantly, heterogeneity is a prominent feature of CF lungs, as infective/inflammatory mucus plaques are not distributed homogeneously throughout the CF lung [18]. 

### 2.3. Disease Comparisons and the Role of Ciliary Beating Versus Mucus Composition 

It has often been questioned whether slowing the “escalator” mechanism of ciliary beating is sufficient to produce CF lung disease [18]. Some hints could be provided by disease comparison with the genetic disease termed primary ciliary dyskinesia (PCD), where the ciliary beating is slowed by a structural defect of the ciliary components without ion transport defects [20]. Hence, CF and PCD represent the paradigm of two pathophysiological mechanisms converging on the same final effect of reducing the ciliary beating frequency, based on defective ion transport or structural ciliary defects, respectively. Interestingly, PCD lung disease is characterized by recurrent infections with *S. aureus*, followed by chronic infections with *P. aeruginosa*, which resembles the lung disease progression in CF. It should not be stated that the clinical status of CF lung disease is more severe than that of PCD due to the wide degrees of disease expression resulting from the genetic mutations causative of CF or PCD; however, if we consider the progression of lung bacterial infection, the average trend of the lung microbiology from PCD patients parallels that of pwCF, but with a relevant delay of years [21]. The small airways are affected in children with PCD at similar rates as in CF, but infections can be cleared more often in PCD than CF [22,23]. This may contribute to the differences in the trend of lung function over time, which can remain stable in PCD patients under regular monitoring and treatment, whereas a progressive respiratory insufficiency is inevitably observed in pwCF [6,24]. Therefore, the similar pattern of infection observed in CF and PCD indicates the key role of the reduction in the ciliary beating frequency in favoring the establishment of chronic bacterial infection. 

Disease progression in pwCF suggests a role for other CF-specific alterations in ASL. Looking for differences, as recalled above, the ion transport defect in pwCF cannot compensate for the increased mucin fraction in the mucus hydro-gel and the incremental secretion of mucin MUC5AC. However, the ASL of individuals with PCD was shown to contain some degree of abnormally increased mucins [25], possibly because of a defective mechanosensitive transduction of the presence of concentrated hydro-gel in the ASL, which does not activate the proper hydration compensatory feedback mechanisms [26]. If mucus abnormalities are a pathophysiological factor as a result of bacterial overload and stasis due to primary structural ciliary defects in PCD [25,26], the opposite sequence of events seems to happen in CF. Consensus is now growing that respiratory mucus alterations have a specific and prominent role in favoring and worsening infection and inflammation in CF lungs. The physicochemical composition of CF mucus reflects the different aspects of the altered ion transport of chloride and bicarbonate. As defective chloride secretion reduces ASL hydration, it favors the polymeric entanglement of mucins which, together with the secretion of mucins MUC5B and MUC5AC, increases the solid fraction of CF ASL almost five times more than that of normal levels [27], making CF mucus characteristically more viscous than normal. The defective bicarbonate secretion into the ASL also contributes to increased mucus viscosity by two proposed mechanisms. As bicarbonate is a chelator of calcium ions secreted from goblet cell granules together with mucins, a high availability of free calcium could result in mucus compaction by increasing the bimolecular bridges promoted by the divalent cation [28]. Alternatively, defective bicarbonate secretion possibly reduces the pH of CF ASL. Although pH reduction in CF ASL is controversial [17,28,29,30], it has been proposed that reduced bicarbonate secretion could favor the protonation of the carboxyl chains of aspartic and glutamic acids, leading to the disruption of the intramolecular salt bridges, causing the mucin to unfold [28]. Indeed, recent findings utilizing bronchoalveolar fluid obtained from CF infants indicated that altered CF mucus is a pivotal defect that precedes bacterial infection [31]. In addition, it has been proposed that altered CF mucus can promote a sterile inflammatory status involving interleukin-1 (IL-1) pro-inflammatory signaling [32] in the early years of pwCF. 

## 3. Bacterial Infections in CF Lung Disease

### 3.1. Microorganisms in Early CF Lung Infections: S. aureus Takes the Stage First

As discussed above, impaired MCC and airway mucus abnormalities promote a favorable milieu for CF lung infections [17,30]. Lung infections in pwCF are polymicrobial and include *Haemophilus influenzae*, *Stenotrophomonas maltophilia*, *Achromobacter xylosoxidans*, *Nontuberculous mycobacteria*, different fungi (e.g., *Aspergillus* and *Candida* species), and respiratory viruses (e.g., rhinovirus, influenza virus, and respiratory syncytial virus). Early in CF, *S. aureus* tends to be the dominant pathogen, while *P. aeruginosa* dominates later in life [33]. A common feature in the recurrent respiratory infections in early life of pwCF is the presence of *H. influenzae* and *S. aureus* in the sputum. However, as these microorganisms are human commensals that are routinely found in the nose or upper respiratory tract, their presence in the sputum brings into question whether there is precedent for a pathologic condition or simply a transient relocation to the sputum. [34]. 

Approaches to *S. aureus* infection are markedly different between the United States (US) and many European and Australian treatment guidelines. Specifically, in the United Kingdom (UK), *S. aureus* antibiotic prophylaxis and the early treatment of any positive cultures is the standard of care, whereas the US guidelines specifically recommend against such treatments [35]. The recommendations for each are based on clinical trials that each showed a reduction in *S. aureus* positive cultures; however, in the US trial, subjects randomized to cephalexin prophylaxis were found to have higher odds of acquiring *P. aeruginosa* [36]. The UK study was smaller and used flucloxacillin and did not indicate increased *P. aeruginosa* rates [37]. These different treatment guidelines may explain the marked differences in the *S. aureus* rates between the countries, with the additional caveat that the UK data only report chronic *S. aureus*, and the US data report any positive cultures. *H. influenzae* is usually quite susceptible to therapy and is transient, whereas *S. aureus* is associated with chronic upper airway colonization or lower airway infection. *S. aureus*, however, is present in the nose of 20–30% of the healthy population. There may be a precedent for *H. influenzae* and *S. aureus* infection in early CF as it was found that even children without CF have a high prevalence for *S. aureus* and *H. influenzae* positive throat swabs [38]. 

In addition to the increased prevalence of *S. aureus* in children, there are other theories that suggest why *S. aureus* is able to cause chronic infection starting in younger pwCF. For example, it has been proposed that the reduced bicarbonate secretion and lower pH of the ASL in CF results in an impaired innate bactericidal activity towards *S. aureus*, possibly by affecting the efficiency of the antimicrobial peptides LL-37 and hBD1 [39]. Additionally, methicillin-resistant *S. aureus* (MRSA) strains occur as chronic lung infections in pwCF with high geographic variations. While in US CF centers, the rate of chronic MRSA is around 25%, it is only 0.4% in Scandinavian countries [40]. The rates of MRSA in CF to some degree reflect those seen in non-CF populations [41]. 

### 3.2. P. aeruginosa and S. aureus Living Together: Cooperation or Competition?

*S. aureus* is commonly detected during childhood, whereas *P. aeruginosa* is less frequently detected in the first years of CF life [33,34]. However, particularly prior to aggressive anti-pseudomonal therapies and prior to CFTR modulators, *P. aeruginosa* becomes the predominant infecting bacterium in pwCF. Chronic infection, marked by mucoid adaptation, is associated with worse clinical outcomes and decreased chances of early eradication [42,43]. Complex and multiple cooperative interactions between these two microorganisms in vitro and in the CF lung in vivo have been reported with partially contradictory results; these studies have been extensively discussed, and we refer the reader to some excellent reviews [44,45,46]. The challenges of in vitro studies include the different adaptation mechanisms that each organism develops during chronic lung infections, the different modes of growth such as biofilm, mucoidy and small colony variants, and the different relative multiplicity numbers that can occur during in vivo infections. The challenges of in vivo studies and epidemiologic data are the growth modes, especially in the small colony variant, such as *S. aureus* auxotrophy, that can be difficult to culture and detect in routine clinical microbiology laboratories. Studies focusing on these adaptive phenotypes have shown rates of 8% [47], 28% [48] and even as high as 42% among clinical isolates of chronic *S. aureus* infection [49]. 

### 3.3. P. aeruginosa Damages the Respiratory Mucosa and Adapts to Persist Chronically

*P. aeruginosa* infection begins in pwCF at a median age of one year. Infection is usually acquired from an environmental source, although person-to-person transmission can occur [50]. Early *P. aeruginosa* isolates typically express an expansive and diverse set of virulence factors that affect the integrity of the respiratory mucosa. Among the surface-bound virulence factors, flagella and pili interact with Toll-Like Receptor (TLR) 5, activating a strong inflammatory response [51]. The Gram-negative endotoxin lipopolysaccharide (LPS) from *P. aeruginosa*, which primarily interacts with TLR4, has been also found to bind CFTR itself [52]. However, the majority of virulence factors are secreted by *P. aeruginosa*. Through the type 3 secretion system, *P. aeruginosa* injects cytotoxins when attached to the host cell membrane, e.g., the apical membrane of bronchial epithelial cells, which the bacterium utilizes to transfer four toxins (ExoS, ExoT, ExoU, ExoY) [53]. This key virulence system has been associated with clinical outcomes and serologic diagnosis in pwCF [54,55,56]. ExoS and ExoT are GTPase-activating proteins that may disrupt the cytoskeleton, making the infected cell “round up”, detach and die [57]. ExoU is a potent phospholipase leading to the loss of plasma membrane integrity [58]. ExoY is an adenyl cyclase that increases the intracellular contents of cAMP and disrupts the actin cytoskeleton [59]. While active during initial infection, the type 3 secretion system is typically turned off as *P. aeruginosa* becomes chronic [60,61]. Quorum sensing (QS) is another virulence mechanism employed by *P. aeruginosa* to coordinate the transcription of about 300 genes by regulatory factors such as RhlR, LasR and PQS, secreting the so-termed “pheromones” [62]. Proteases are key virulence factors regulated by QS [63]. In this protease repertoire, a key role has been attributed to the protease LasB, which was found to reduce the expression of CFTR and subvert the host immune response [64,65]. Pyocyanin, a bacterial metabolite of the class of phenazines, produces direct cell damage and a reduction in ciliary motion [66] and, specifically for CF, reduces the expression of the CFTR protein [67]. Exotoxin A, released by the type 2 secretion system, blocks protein synthesis by the ADP-ribosylation of the Elongation Factor 2, producing mitochondrial dysfunction and DNA degradation, thus leading the cell to apoptosis [68,69]. The epoxide hydrolase termed CFTR Inhibitory Factor (Cif), secreted by *P. aeruginosa* and delivered by the bacterial outer membrane vesicles to the bronchial epithelial cells [70,71,72,73], acquires a prominent role in CF because of its effect on the degradation of the CFTR protein, achieved by increasing its ubiquitination and proteasomal degradation [74]. A wider number of nonspecific proteases, phospholipases and hydrolases are also secreted by *P. aeruginosa*, and these have been discussed in excellent reviews [75,76,77].

Despite being active during early infection, the virulence genes are usually turned off as the infection becomes chronic. This long series of virulence traits are potent modifiers and destroyers of the bronchial epithelial cells, although it is clear that *P. aeruginosa* does not express all of them during the whole evolution of its interaction with the airway mucosa. Upon early intermittent infection periods in CF lungs, planktonic *P. aeruginosa* is subjected to a selective pressure in the milieu of the bronchial ASL. Different conditions try to eliminate the planktonic *P. aeruginosa* from the surface of the CF mucosa. Strategies by the host to eliminate *P. aeruginosa* likely contribute to this adaptation to chronicity and the selection of certain phenotypes. Despite the advantage of the defective “escalator” function of the MCC and the altered mucus composition in CF, planktonic *P. aeruginosa* survival should face phagocytosis by neutrophils and macrophages, the presence of antibacterial peptides and antibiotic drugs at high doses trying to eradicate the infections, as well as Reactive Oxygen Species (ROS) released by activated neutrophils [78,79,80,81]. In particular, the exposure of *P. aeruginosa* to ROS is thought to represent the principal mechanism that induces damage in the bacterial genomic DNA, leading to the mutations implicated in the persistence of infection [82,83,84,85,86]. However, as we discuss in the next section, the interactions of *P. aeruginosa* with the innate immune system often fail to control the infection and may inadvertently worsen the situation. 

## 4. *P. aeruginosa* Cross-Talk with CF Airway Mucosa

### 4.1. A Sticky Situation: P. aeruginosa Biofilms and the CF Airway Epithelium

In the CF airway, during chronic infection, *P. aeruginosa* resides within the stagnant mucus plugs and plaques; histological studies rarely find colonization or direct physical interaction with the airway epithelium [87,88]. The accumulated, dehydrated mucus provides a scaffold in which the bacteria preferentially colonize and form biofilm-like structures. A recent study found that the macro- and micro-rheological properties of CF-like sputum render *P. aeruginosa* biofilms highly tolerant to chemical and mechanical stresses [89]. Given that the mucin concentration typically increases with age [90], it is not surprising that *P. aeruginosa* infections also become more difficult to treat with time. While there are many factors driving the temporal increase in the mucin concentration, *P. aeruginosa* infection both exacerbates and exploits CF disease progression. During early infection, *P. aeruginosa* is typically highly motile [91]. Flagellin, the major structural subunit of flagella, induces the expression of MUC5AC and MUC2 when exposed to the airway epithelial cells in a TLR5/NAIP-dependent manner [92], and likely contributes to the early goblet cell hyperplasia and hyper mucus production that further contributes to CF pathology, creating an even greater opportunity for *P. aeruginosa* biofilm infections to spread. 

Despite being a biofilm infection that primarily resides in the mucus, *P. aeruginosa* still has a profound impact on the airway epithelium. Although *P. aeruginosa* typically loses its acute virulence factor expression as the infection becomes more chronic, there are still some virulence factors that contribute to CF pathology throughout infection. Pyocyanin and elastase play significant roles in airway epithelial damage. Elastase is more prominent during early infection [93] and was found to decrease both the residual CFTR activity and expression, making it a direct contributor to worsening infection and CF pathology. Further, elastase also degrades several immune mediators, including interleukin-6 (IL-6) [64]. IL-6 degradation results in the dysregulation of the IL-6/STAT3 signaling pathway and prevents the IL-6-mediated repair of the epithelium [64]. Saint-Criq et al. also showed that IL-6 was important for controlling acute *P. aeruginosa* lung challenge, suggesting a highly important role of IL-6 signaling during early infection [64]. Pyocyanin also likely plays a significant role in transforming the airway to promote chronic infection. Some of the earliest studies of pyocyanin and the CF airway found that pyocyanin inhibits ciliary beating and significantly decreases the mucus velocity in the airway [94], implicating pyocyanin in the creation of the chronic infection environment that *P. aeruginosa* thrives in. A more recent study provided further evidence implicating pyocyanin in shaping the chronic infection environment. Mucus accumulation and stagnation in the airways is a hallmark of CF. However, there are also changes to the mucus polymers that may facilitate infection. Jeffries et al. found that pyocyanin modulates mucin glycosylation by altering the expression of two glycosyltransferases in the CF airway epithelial cells. This alteration led to an enhanced sialylation of mucins, which promotes *P. aeruginosa* binding, and presumably further promotes infection [90,95,96,97]. 

Metabolism also plays a key role in *P. aeruginosa* infection in CF. Chronic *P. aeruginosa* infection of the CF airways is associated with a significant shift in the host metabolism [98,99,100,101,102]. This metabolic shift may drive the characteristic biofilm lifestyle of *P. aeruginosa.* In addition to mucus dehydration and stagnation, CFTR dysfunction also leads to changes in the host cell metabolism, particularly in immune cells. The tumor suppressor gene, PTEN, relies on CFTR for positioning at the plasma membrane [103]. Riquelme et al. found that defective CFTR results in impaired PTEN positioning and, therefore, impaired downstream signaling of the PI3K-Akt pathway, which increases the susceptibility to *P. aeruginosa* challenge [103]. In a follow up study, the same group found that the combined CFTR and PTEN dysfunction resulted in the aberrant stimulation of mitochondrial activity and the release of succinate [104]. As succinate is the preferred carbon source for *P. aeruginosa* [105,106,107,108], it has been postulated that the release of succinate further aids *P. aeruginosa* infection [103,109]. In addition, *P. aeruginosa* infection stimulates the release of itaconate from the host immune cells [110]. It was found that host-derived itaconate induces biofilm formation, which is favorable to *P. aeruginosa* in the hyper mucus environment. Itaconate induces bacterial membrane stress [111], which was found to suppress LPS assembly and also to increase the expression of alginate biosynthesis genes, resulting in increased biofilm formation. 

### 4.2. Dysfunctional CFTR-Driven Lung Inflammation

Inflammation begins early in CF life, potentially before birth [112], and infection-free lung inflammation has been found in the bronchoalveolar lavage fluid of CF infants [113]. That inflammation precedes infection in the lungs of pwCF has been further corroborated by different studies, including some conducted with CF airway xenografts [114]. Both murine and porcine animal models of CF further support the role of the inherent CF pathology in generating inflammation without infection. Using a porcine model of CF, it was found that, even from birth, CF pigs possessed higher bacterial burdens in the lungs [115,116]. The use of antibiotics from birth was sufficient to control the bacterial burden; however, by three weeks of age, the CF pigs still exhibited significant inflammation, similar to that of CF pigs raised without antibiotics. This suggests that early airway inflammation subsists even when the bacterial infection is removed from the equation. Studies using the CF-like *Scnn1b*-transgenic mouse model found that there was no difference in the mucus obstruction between conventional or gnotobiotic mice [117]. More importantly, it was found that gnotobiotic mice still developed significant airway inflammation, further supporting the role of CF disease in promoting the inflammation that may ultimately set the stage for pathogen colonization [117]. These and many other findings opened a still ongoing debate on sterile lung inflammation in pwCF, which is beyond the scope of this article, but has been discussed in excellent reviews by others [118,119]. The evaluation of the different cellular components of the CF airways mucosa has been performed to understand the role of defective CFTR on the innate and adaptive immune functions. Based on the studies in pwCF and CF animal models, CF could be devised as a peculiar mucosal immunodeficiency paradigm [120]. 

Early in CF disease, the airway epithelial cells lining the bronchial walls can experience hypoxia due to mucus plugging and elevated oxygen consumption, leading to cell stress and necrosis [121]. This early structural damage is associated with higher rates of respiratory decline and airway infection, primarily with *S. aureus* and/or *P. aeruginosa* [122,123]. It was also found that structural damage to the lung was not required for the onset of inflammation, but rather that mucus accumulation and inflammation both precede structural lung damage and infection [31]. Significant constituents of pre-infection inflammation are IL1-β and IL1-α, which were found to induce a positive feedback loop of mucus overproduction through the stimulation of the downstream signaling pathways including SPDEF and ERN2 [124]. Indeed, the endoplasmic reticulum (ER) stress transducer Inositol Requiring Enzyme 1β (IRE1β), the protein encoded by *ERN2*, and that has been implicated in airway mucus overproduction, is up-regulated in CF bronchial epithelia [125], indicating a role for IRE1β in CF airways disease. 

Early work with CF neutrophils questioned how the inherent CFTR defect in the airway epithelium may impact the neutrophil function, apart from the accumulated mucus, making it difficult for the neutrophils to reach the pathogens. It was found that neutrophils also express CFTR and that CFTR dysfunction impacts the ability of neutrophils to respond to pathogens, despite only having a low basal expression of CFTR. CFTR-deficient neutrophils poorly clear *P. aeruginosa*, leading to extended infection and greater mortality in mice [126,127]. Sputum neutrophils from pwCF were also shown to have reduced respiratory burst and higher levels of necrosis [128]. However, it is still unknown how much of this “functional exhaustion” [128] is a result of underlying CF disease or the unrelenting, nonstop battle against invading pathogens.

CF macrophages exhibit reduced efficiency in removing dead neutrophils [129], which is a key process for reestablishing tissue homeostasis. Several studies have indicated that macrophages play a critical role in the defective CF host response to *P. aeruginosa* by sustaining a proinflammatory status via preferential polarization to a M1 phenotype [129,130,131,132,133,134]. The M1 polarization of macrophages is mainly applicable to in vitro investigations, with limited translation into an in vivo setting [135]. However, a possible consequence of CF macrophage dysfunction and M1 polarization has been suggested regarding an imbalance in the anti-inflammatory regulation of the adaptive immune branch, where the presence of reduced Tregs cells has been observed in CF patients, accompanied by preferential increments of Th-17 cells (the latter are actively involved in neutrophil recruitment in CF bronchi [136,137,138], an imbalance that does not seem corrected by CFTR modulators [138].

Notably, the CF mucosal immunodeficiency appears strictly pertinent to the lung, as the dysfunction of CF phagocytes is not systemic. Whether the impaired function of the CF immune cells in the lungs is directly linked to the basic CFTR defect or resides downstream in the pathological cascade of secondary events [139] will require further studies. Nevertheless, by the time pwCF encounter *P. aeruginosa*, the airway environment is already in a hyper-inflamed state that likely favors colonization and chronic infection.

### 4.3. Inordinate Inflammation: P. aeruginosa Adds Fuel to the Fire

*P. aeruginosa* further exacerbates inflammation in the CF airway. A longitudinal study by Jaudszus et al. found that both IL-8 and IL-6 were significantly increased in pwCF newly colonized with *P. aeruginosa* [140]. Further, in a large cohort study including over 100 pwCF, it was found that pwCF whose lungs were predominantly infected by *P. aeruginosa* had higher levels of IL-8, IL-1β and TNF-α compared to individuals that were predominantly infected by *S. aureus* or *Achromobacter xylosoxidans* [141]. A deeper investigation into the mechanism behind how *P. aeruginosa* induces the high expression of IL-6 and IL-8 in CF bronchial epithelial cells revealed that IL-6 and IL-8 were induced through the activation of p38 MAPK and Syk kinase [142]. The inhibition of p38 MAPK and Syk kinase prevented the release of IL-6 and IL-8. Additionally, during the normal course of infection in early CF, where *P. aeruginosa* typically still possess flagellar motility, flagellin monomers are naturally shed. It was found that the airway epithelium can endocytose flagellin monomers and further induce pro-inflammatory signaling [143]. The endocytosis of flagellin induced TNF-α and IL-6 cytokines, as well as the CXCL1, CXCL2, CXCL3 and CXCL20 chemokines [143].

Another key part of *P. aeruginosa* infection is the genotypic and phenotypic diversification that *P. aeruginosa* undergoes during chronic adaptation [144,145]. The acquisition of a mucoid phenotype, or isolates that hyperproduce exopolysaccharides, is one such adaptation in CF. A recent study by Malhotra et al. investigated the regional differences in *P. aeruginosa* isolates and inflammation. While mucoid and non-mucoid isolates were not spatially separated, regions dominated by mucoid isolates were associated with higher IL-1β, IL-6, IL-8 and TNF-α [146]. The increasing inflammation over time in pwCF infected with *P. aeruginosa* is likely a result of the conversion to mucoid isolates also observed over time. As such, early intervention strategies that aim to prevent the aberrant inflammation, even prior to pathogen colonization (i.e., from birth), represent a key step in fighting not only the underlying CF disease, but also the invading pathogens. Another common adaptation seen in chronic *P. aeruginosa* isolates is modifications in the structure of LPS. The most common change in LPS seen in chronic isolates is the acylation of lipid A [147,148]. Counter intuitively, however, this heavy acylation increases the recognition of LPS by TLR4 and stimulates NK-κB activation, which further worsens the inflammation [149,150]. However, enhanced TLR4 recognition may be a tradeoff for an increased resistance to cationic antimicrobial peptides [151]. 

### 4.4. Frustrated Phagocytes: The Battle of Neutrophils against P. aeruginosa

Phagocytes such as neutrophils and macrophages represent some of the first-line defenses against respiratory pathogens. During typical airway infection, pathogens are trapped in the healthy mucus, where they are met by resident alveolar macrophages, which initiate the innate immune defense [152]. Neutrophils are then recruited and quickly destroy the invaders. However, in CF, neutrophils represent the primary immune cell associated with the response to infection. Numerous reports show that neutrophils are usually highly effective in controlling the challenge with *P. aeruginosa*, and that neutropenia results in an increased susceptibility to *P. aeruginosa* [153,154,155,156]. In CF, the phagocyte-pathogen dynamics shift dramatically in favor of the pathogen. Even prior to significant pathogen colonization, there is significant pre-existing neutrophil infiltration. However, despite the robust recruitment of neutrophils, the pathogen clearance is severely diminished. This is likely due to a combination of host deficiencies, as well as an inherent or acquired resistance to phagocytes by *P. aeruginosa*. 

Despite the handicap, neutrophils still have the means for controlling *P. aeruginosa* infection. As *P. aeruginosa* infection is primarily a biofilm infection, traditional phagocytosis is ineffective. Instead, neutrophils rely heavily on degranulation and the release of neutrophil extracellular traps (NETs). Neutrophils that undergo NETosis also release large amounts of chromosomal DNA as part of the NETs, which may ultimately benefit *P. aeruginosa* [89]. Primary azurophilic granule components. such as neutrophil elastase (NE), myeloperoxidase (MPO) and other antimicrobial effectors. represent some of the most potent defenses that neutrophils employ to destroy bacteria. Indeed, both NE and MPO are present in high concentrations in CF sputum [157,158]. While MPO is usually sufficient to kill ingested bacteria, the efficiency of MPO to kill extracellular bacteria through NETs is unknown, but the efficiency likely drops dramatically, and its efficacy is diminished in CF [159]. 

Antimicrobial peptides such as azurocidin, LL37 and the bactericidal/permeability increasing protein (BPI) also represent defenses that have been shown to be effective against *P. aeruginosa.* BPI sputum levels were found to be associated with *P. aeruginosa* infection in pwCF and were also found to be critical for the clearance of *P. aeruginosa* in a murine model [160,161,162]. The killing of *P. aeruginosa* by BPI is mediated through its ability to bind bacterial LPS, which causes perturbations to the outer membrane of the Gram-negative bacteria, leading to membrane blebbing and destabilization, finally resulting in bacterial cell death [163,164]. 

In contrast to azurocidin and BPI, which are neutrophil granule specific antimicrobial peptides, LL37 is also released from the epithelial cells [165]. Similar to BPI, LL37 binds to the outer membrane of the Gram-negative bacteria, leading to bacterial cell death [166]. Interestingly, it was found that LL37 did not kill laboratory strains of *P. aeruginosa*, but was found to efficiently kill clinical isolates [166,167]. It was also found that pathogens with a more neutral membrane charge were more resistant to LL37 compared to *P. aeruginosa* [167]. The differential killing of the laboratory strain compared to the clinical isolates may be related to the modifications of lipid A that occur in the CF airway [148,151,168], although more investigation is needed to delineate the differential killing effects of LL37. 

NE is a serine protease, and one of the most abundant granule and NET effectors. NE has many roles in infection, including direct antimicrobial action, as well as being a regulator of NETosis. Once active, NE can be released in the primary azurophilic granules in order to battle infection, but is also involved in the production of NETs. During NETosis, the neutrophilic chromatin decondenses and is eventually released into the environment upon the lysis of the cell. NE has been shown to be important for the degradation of histones, leading to the decondensation of the chromatin [169]. 

In terms of its antibacterial activities, NE targets outer membrane proteins to kill bacteria. In *P. aeruginosa*, it was found that NE can degrade the outer membrane protein, OprF, which leads to outer membrane destabilization and bacterial cell death. Another identified target of NE is *P. aeruginosa* flagellin [170,171]. Although NE is efficient in controlling pathogens in acute infection, and not in the context of mucus obstruction, NE continually worsens infection in muco-obstructive airway diseases. NE likely aids *P. aeruginosa* to establish a more chronic infection based on the observations of the loss of or mutation of the two main targets of NE. It was found that mutants of OprF exhibited enhanced biofilm forming capabilities and the production of exopolysaccharides [172]. Additionally, in traditional biofilm development, while flagella are required for the initial stages of biofilm formation, they are ultimately downregulated and turned off as the biofilm matures. In conclusion, the NE targeting of flagellin and OprF likely aids *P. aeruginosa* chronicity, and further contributes to the vicious cycle of infection and inflammation.

Another important aspect to most bacterial infections, especially chronic infections, is the battle for nutrients, including transition metals such as zinc, manganese and iron. One of the most successful strategies employed by the host to combat bacterial pathogens is nutritional immunity, and transition metals represent one of the biggest targets of nutritional immunity. Iron, zinc, manganese and copper are essential cofactors for many enzymes, especially those involved in the SOS response, DNA repair and the oxidative stress response. As such, the sequestration of these metals by the host is a highly effective strategy in controlling infection. The most abundant metal sequestering effectors employed by neutrophils are calprotectin (CP) and lactoferrin (LF). Calprotectin is primarily a zinc and manganese chelator, while lactoferrin is an iron chelator. During neutrophil activation and NETosis, CP comprises up to 60% of the soluble cytosolic protein content and is present in milligram quantities in CF sputum [173,174,175]. CP is constitutively expressed at relatively low levels. In instances of host cell damage, such as neutrophil or epithelial cell damage, calcium is released into the environment. Once CP binds calcium, its affinity for zinc and manganese rises dramatically [176]. However, despite CP’s potent activity, *P. aeruginosa* does have its own zinc chelation system that rivals CP. The *P. aeruginosa* metallophore, pseudopaline, has been identified as a potent zinc chelator and is able to acquire zinc in the presence of CP [177,178]. 

The role of CP extends beyond the chelation of zinc or manganese. It was found that CP enhanced the interaction between *P. aeruginosa* and *S. aureus* and aided in promoting co-colonization in a mouse model [179]. As such, CP and metal fluctuations have significant impacts on the community structure and polymicrobial interactions. 

The battle for metals extends to iron, as neutrophils release the iron chelator lactoferrin during degranulation and NETosis. Lactoferrin has been shown to be a potent inhibitor of the biofilm development of *P. aeruginosa* [180,181]. Aerosolized lactoferrin has been shown to be a promising treatment option, based on studies in a CF mouse model. Aerosolized bovine lactoferrin reduced the bacterial load in the lung, as well as decreased the infiltrating neutrophils [182]; these studies further indicate that lactoferrin is a potent anti-*Pseudomonas* effector. *P. aeruginosa*, however, is not without its own defense against lactoferrin. *P. aeruginosa* possesses a variety of siderophores, including pyoverdine and pyochelin. However, it was found that pyoverdine production actually decreases over time as the infection becomes more chronic [183,184]. Instead, systems associated with heme utilization and iron sequestration from hemoglobin were found to be the primary method for iron acquisition during late chronic infection [183,185,186]. There is also precedent for the availability of ferrous ions in CF sputum; in one study, the high expression of the FeoB ferrous ion transporter was the only iron acquisition system detected in sputum [183]. The switch to heme utilization and the uptake of free iron may be a result of progressive damage to the airway. The degradation of lactoferrin by the host proteases likely frees the iron, and damage to the epithelium likely releases high concentrations of heme into the environment, thus prompting *P. aeruginosa* to change its iron acquisition strategies. 

## 5. Post CFTR Modulator Changes in Airway Microbiology

### 5.1. Overview of CFTR Modulators and Clinical Impact

The new treatments addressing the underlying defect in CF, namely the function of CFTR, have led to a new era of CF care for those eligible for such therapies. The small molecule compounds known as “potentiators” enhance the channel gating of the CFTR protein, whereas the compounds known as “correctors” enhance CFTR folding and trafficking (in the case of F508del or other mutations with impaired folding), leading to better CFTR expression at the apical membrane. The initial compound, ivacaftor, which was directed towards the G551D mutation, led to improvements in lung function, weight gain and a decline in the loss of lung function over time [187,188]. There have been subsequent efforts focused on the F508del mutation, which is present in more than 80% of pwCF. A triple combination of two correctors (elexacaftor and tezacaftor) with the potentiator ivacaftor (ETI) has shown to have significant clinical impact. with improvements in FEV_1_ of around 10–15% [189,190]. Based on those clinical successes, ivacaftor and ETI are often referred to as “highly effective modulator therapy” (HEMT), as other modulators are less beneficial in vivo. These clinical studies included pwCF with mild to moderate disease, as measured by lung function. The subsequent post marketing real-world data showed tolerance and significant effects in those with severe disease, including a reduction in lung transplant referrals [191,192]. Notably, these therapies also improve non-pulmonary outcomes. such as chronic sinusitis, nutrition and CF-related diabetes [193,194], which is indicative of their effect being due to improvement in the CFTR function. 

However, around 10–20% of pwCF are not eligible for these CFTR therapies, based on their CFTR mutation, age or not tolerating the therapies. In addition, 10% or more pwCF cannot benefit from HEMT because of mutations leading to reduced CFTR mRNA and protein levels resulting from severe splicing or nonsense CFTR mutations [195,196]. Indeed, stop mutations and severe splicing mutations in the CFTR gene result in reduced channel activity [196], severe CF disease phenotypes [197] and resistance to HEMT [195,196]. This rate of not qualifying is also affected by patients’ genetic background, country of residence and insurance. There is a strong focus for ongoing research to identify compounds, test them in vitro and evaluate them in clinical studies.

### 5.2. Effects of CFTR Modulators on Airway Mucus, Inflammation and Bacteria

The main effect of HEMT is the improvement in the CFTR function, by ~50% in in vitro models, with different in vivo effectiveness being based on tissue and the other concomitant CFTR mutations. The in vitro evaluation and improvement of transepithelial conductance is used to assess the CFTR function, with nasal cells being used to allow for the personalized testing of modulators for pwCF with uncommon mutations [198]. Based on the close correlation of those measures to the improvement of the CFTR function in vivo, these assays have been accepted by the Food and Drug Administration (FDA) for clinical eligibility [199]. Ivacaftor has also been used in-utero and during postnatal treatment in ferrets harboringthe G551D CFTR mutation [200]. The findings indicate that early therapy prevents perinatal disease and the maintenance of therapy is necessary to prevent disease reoccurrence. The effect of ivacaftor has also been tested on G551D humanized rats regarding age-dependent abnormalities of the airway mucus [201] and inflammation [202]. Additional animal studies also confirm that MCC improves with CFTR modulator therapy. For a current review of CF animal models in the era of HEMT, the reader is directed to [203]. With an improvement in the CFTR function, fluid and electrolyte transport across the airway epithelia is closer to normal function. Studies in primary bronchial epithelial cultures, CFTR-knock out cell lines and a CF rat model showed an improved hydration of the periciliary liquid and reduced mucus viscosity [201,204,205]. Increases in MCC have also been demonstrated in pwCF with G551D mutation who initiated ivacaftor. In particular, clearance from the peripheral lung compartments was significantly improved at one month and maintained at three month examinations [206]. The authors also noted that additional effects, in addition to MCC, e.g., on inflammatory cells, may contribute to the clinical outcomes. These effects help to restore or improve MCC to combat infections. As discussed below, the propensity to infection could also be affected by the effects on the immune cells. It is noteworthy that additional ion channels are functionally relevant and are being evaluated to improve the hydration of mucus [207]. To date, the effects of HEMT on inflammatory responses in clinical studies are still variable, likely due to differences in the pre-existing disease severity e.g., irreversible bronchiectasis, with some of the patient groups studied having ongoing infection [208] and because of the limited consensus on the clinical parameters to reliably assess the effectiveness of drugs on CF lung inflammation [209]. 

Several groups have evaluated the direct antimicrobial activity of ivacaftor on the grounds of its structural similarity to quinolones. Using *S. aureus* and *P. aeruginosa* laboratory strains, one group showed some efficacy against *S. aureus* but little efficacy against *P. aeruginosa* [210]. Another study, using clinical isolates, showed the potentiation of a bactericidal effect of tobramycin on *S. aureus*, but no activity against *P. aeruginosa* [211]. The effectiveness of ivacaftor against *S. aureus,* but not *P. aeruginosa*, may be explained in that *P. aeruginosa* possesses the quinolone-based QS system, PQS, which produces several anti-staphylococcal compounds, such as the respiration inhibitor, HQNO [45]. 

A recent study that investigated the ability of HEMT and antibiotics to synergize found minimal effects. It was found that ETI treatment did not impact the efficacy of most antibiotics against *P. aeruginosa* clinical isolates, with the exception of polymyxin B. Interestingly, the additive effect of ETI with polymyxin B was also seen in ivacaftor alone-treated isolates, indicating that elexacaftor or tezacaftor are devoid of additional synergistic abilities for HEMT [212]. Further, the effect of HEMT on the polymyxin B efficacy against *P. aeruginosa* may not be related to any direct effect on the bacteria, but may be a result of the mucus clearance and reduced mucus content in the airways, as the polymyxins are known to be inhibited by mucins due to the charge-charge interactions. Thus, it still remains to be seen how HEMT directly influences *P. aeruginosa* over time.

Studies regarding ETI therapy and its role in directly influencing bacterial infection, particularly *P. aeruginosa* and *S. aureus*, are still in their infancy, with many longitudinal studies forthcoming. Instead, the majority of HEMT’s influence on infection is most likely a result of improved immune function and mucus clearance. Single-cell RNA sequencing studies have characterized sputum immune cells from pwCF and healthy subjects. pwCF had predominantly recruited monocytes and neutrophils, with three types of monocytes and neutrophil subpopulations, indicating an immature proinflammatory CF airway status [213]. These findings have contributed to the understanding of CF immune dysfunction for CF airway pathogenesis; moreover, they indicate the evaluation of immune cells in sputa as a feasible approach to test the clinical effects of HEMT on the immune cell responses. Indeed, previous studies have suggested that CFTR modulators may also affect the host immune responses. Hisert et al. reported that ivacaftor changes the transcriptional profiles of blood monocytes, increases the plasma concentrations of the chemokines CXCL2 and CCL2 [214] and decreases the blood monocyte sensitivity to IFN-γ [214]. Studies evaluating the function of monocytes and macrophages in pwCF before and after the initiation of ETI have shown that the phagocytic function and intracellular killing improved on ETI therapy [215,216]. These changes were associated with improvements in the clinical outcomes (lung function and BMI) [215,216]. Studies in patients initiating ivacaftor also examined peripheral blood monocytes using single-cell transcriptomic changes pre- and post-ivacaftor therapy. Changes in the genes related to cell differentiation, microbial infection, inflammation and Toll-like receptor signaling were seen [217]. Although the study highlights the effects of CFTR modulators on the immune cells, it cannot answer the question of the direct effects vs. secondary changes due to an improvement in the end organs. Notably, the frequency of positive cultures for CF pathogens decreased; however, because sputum production was decreased after ETI initiation, this could have contributed to the lower detection rates. A current review on the effects of HEMT on CF microbiology and inflammation can be found at [218].

### 5.3. Effects on Pulmonary Infection–Clinical Evidence

CF patient registries are available for the US, Canada and most European countries, and allow us to assess the real world outcomes with these new medications. A comparison of patients on (n = 276) vs. not taking ivacaftor (n = 5296) in the UK showed a reduction in the *P. aeruginosa* prevalence ratio, of 0.68 (95% confidence interval, 0.58–0.79; *p* < 0.001), but no significant changes for *S. aureus*. Importantly, the rate of *P. aeruginosa* acquisition was also lower for those on ivacaftor (36.6% vs. 48.6%); however, those eligible for ivacaftor were younger and had milder lung disease at baseline, consistent with the genotype that made them eligible for ivacaftor [219]. 

Several smaller studies evaluating the microbiological outcomes in individuals eligible for ivacaftor showed a 29% reduction in *P. aeruginosa* positive cultures, but no significant decrease in *S. aureus* [220]. Other studies have evaluated the changes in respiratory microbiome at different time points. For instance, a decreased bacterial load was reported by one group at one month of therapy, while no change in the bacterial load was observed in two studies at six and 12 months, respectively [221,222]. Differences were noted based on the severity of the disease and the prior duration of infection.

Real life data for ETI are more difficult to assess as the introduction of ETI in the US coincided with the onset of SARS-CoV2, which led many pwCF to only seek minimal care at CF centers, which reduced the number of cultures conducted. Several prospective clinical studies, however, are evaluating these issues, such as the PROMISE (https://clinicaltrials.gov/ct2/show/NCT04038047 accessed on 30 October 2022) and the RECOVER (https://clinicaltrials.gov/ct2/show/NCT04602468 accessed on 30 October 2022) studies. These indicate a reduction in the number of positive cultures for *P. aeruginosa* and, to a lesser degree, for *S. aureus*, but mostly show significant improvements in the markers of CF disease, i.e., a reduction in sweat chloride, sustained improvements in lung function and weight gain [223].

## 6. Impact of Chronic Infection and Inflammation on CF Airway Epithelia

### 6.1. Adaptive Airway Epithelial Response to Chronic Infection and Inflammation

The chronic exposure of the CF airway epithelia to the infectious/inflammatory airway milieu induces key alterations in the epithelial structure and function. For instance, earlier studies demonstrated that the short-term (6–11day old) primary cultures of F508del/F508del CF human bronchial epithelia (HBE) exhibit a hyperinflammatory phenotype characterized by increases in the baseline and bradykinin-stimulated IL-8 secretion when compared to short-term primary cultures of normal HBE [224]. Notably, the hyperinflammatory responses of primary F508del/F508del HBE cultures were lost in long-term (30–40 day old) cultures, indicating that they were not associated with the F508del CFTR mutation [224]. In addition, the hyperinflammatory phenotype could be restored by luminally exposing the long-term CF HBE cultures or long-term normal HBE cultures to supernatant from mucopurulent material (SMM) from human CF airways [224]. These findings indicated that the hyperinflammatory state of short-term CF cultures was an adaptive response to the chronic infectious/inflammatory environment of the CF airways in vivo, which was lost under prolonged culturing conditions in vitro.

### 6.2. Relevance of Endoplasmic Reticulum (ER) Ca^2+^ Store Expansion to Inflamed Normal and CF Airways

The hyperinflammatory responses of short-term CF cultures or long-term CF cultures exposed to SMM were mediated by the expansion of the apically-localized ER Ca^2+^ stores, based on (1) functional data (e.g., increased apical bradykinin- or UTP-promoted ER Ca^2+^ release) and (2) morphological evidence (e.g., higher apical expression of ER Ca^2+^ store markers) [224,225]. The expansion of the ER Ca^2+^ stores was also observed in native F508del CF HBE [224,225], confirming its relevance to the CF airways pathophysiology. Importantly, the ER Ca^2+^ store expansion of the inflamed CF airway epithelia can amplify Ca^2+^-mediated cytokine production and, thus, contribute to the robust inflammatory status of CF airways [224].

An additional important consequence of the expansion of ER Ca^2+^ stores in chronically inflamed CF airway epithelia is its possible impact on the mitochondrial function. In airway epithelia, the ER and the mitochondria physically interact, and their proximity facilitates an efficient mitochondrial Ca^2+^ uptake following the release of Ca^2+^ from the ER sites [226]. We speculate that the increased mitochondrial Ca^2+^ uptake in response to the higher levels of Ca^2+^ released from the expanded ER network may have functional consequences for CF pathophysiology. For instance, higher mitochondrial Ca^2+^ uptake can lead to an increased Ca^2+^-dependent mitochondrial production of ROS. Indeed, it has been proposed that mitochondria-associated ER membranes (MAMs) can regulate the inflammatory processes [227,228] and the mitochondrial Ca^2+^ uniporter has been suggested as a novel therapeutic target for CF lung inflammation [228].

We note that the ER Ca^2+^ store expansion should be beneficial for infected normal airways as the increased Ca^2+^-dependent inflammatory responses, together with a normal MCC, would help to clear the airways from the infectious insult [224]. However, in CF airways, the amplified Ca^2+^-mediated inflammatory responses might be inefficient in clearing the infection due to the presence of airway muco-obstruction [224,229,230]. Hence, this beneficial innate defensive mechanism becomes maladaptive in CF airways, further contributing to the robust inflammatory status of CF airways and the promotion of airway wall damage.

### 6.3. The CF Airway Inflammatory Milieu Enhances CFTR Rescue

Another consequence of airway epithelial inflammation is its impact on the folding and expression of proteins, including ER chaperones [224,230,231,232]. These airway epithelial alterations may alter the efficacy of CFTR correctors acting at the ER compartment. Although airway inflammation is present in pwCF at the initiation of HEMT, until recently, most in vitro studies evaluating CFTR rescue have not utilized the inflammatory conditions relevant to CF airways in vivo. Utilizing the SMM model to inflame primary F508del homozygous HBE cultures, it was found that SMM enhanced the lumacaftor-increased F508del CFTR-mediated responses to forskolin and VX-770; these functional data correlated with an augmentation of lumacaftor-rescued band C (mature) CFTR in SMM-exposed cultures [233]. In addition, SMM exposure overcomes the chronic ivacaftor-inhibited F508del rescue by lumacaftor or tezacaftor [233,234]. The enhancement of CFTR rescue by inflammation is not limited to these modalities of modulator therapies, as SMM-promoted inflammation also increases the F508del CFTR rescue resulting from various combinations of modulators, including a triple combination [234]. These studies demonstrated that CF airway epithelial inflammation enhances F508del CFTR maturation and facilitates CFTR trafficking to the apical membrane, thereby augmenting the functional response of apically-localized F508del CFTR. These findings, which have led to the new concept that the efficacy of HEMT is enhanced by the inflammation of the CF airway epithelia [233,234,235], have clinical implications. Notably, recent studies by Rehman et al. have indicated that inflammation enhances HEMT efficacy in vitro and in vivo [236]. Their in vitro studies utilized TNFα and IL-17 as inflammatory stimuli, and their findings are in agreement with the above in vitro studies indicating that CF airway inflammation enhances the efficacy of CFTR modulators. Rehman et al. also found a positive correlation between airway inflammation and the improvement in lung function in pwCF receiving ivacaftor [236]. These initial studies are in agreement with the notion that the CF airway milieu also has a positive impact on the efficacy of HEMT in vivo. These findings suggest that the impact of anti-inflammatory treatments should be considered in the CF clinic for pwCF receiving HEMT. However, IL-17/TNFα have been recently shown to produce hyperviscosity, affecting mucociliary clearance [237], and other studies have reported that the inflammatory mediator TGF-β decreases the CFTR function in the airway cells of non-CF subjects [238] and CFTR biogenesis [239], and negatively affects F508del CFTR rescue [239]. Additional studies will be necessary to address the mechanism for inflammation-enhanced CFTR rescue and the interplay between anti-inflammatory therapies and HEMT efficacy. In parallel, the real effect of HEMT on the inflammatory process damaging the airway tissue of adult pwCF is under scrutiny, as the criteria to evaluate the effectiveness of drugs on CF lung inflammation in clinical trials are contradictory and require a new consensus [240].

## 7. Conclusions

The past several years have brought significant progress to CF basic science, translational and clinical research. New treatments targeting the CF mucus alterations have been developed; a vast amount of knowledge has been acquired regarding the CF microbiome and how it is altered throughout the course of the disease; and the use of HEMT has been revolutionary for most pwCF. However, because pwCF with rare CFTR mutations are less responsive to the available HEMT, there is an unmet need for novel therapies for these pwCF. Further improvements in mucus-targeting therapies are also needed, as well as the development of novel antimicrobials and anti-inflammatory drugs. Additional studies are necessary to address how the CF airway inflammatory milieu enhances HEMT efficacy and whether anti-inflammatory drugs decrease the clinical efficacy of HEMT.

Future research is expected to increase our understanding of CF pulmonary pathophysiology, the impact of long-term HEMT on CF airway infection and inflammatory status, and lead to the development of novel therapies for pwCF with rare mutations. The future of CF research will continue to offer many challenges as well as many rewards.

## Data Availability

Not applicable.

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
