# Peer review of "Revisiting Host-Pathogen Interactions in Cystic Fibrosis Lungs in the Era of CFTR Modulators"

_ijms, 2023, doi:10.3390/ijms24055010_

Round 1
Reviewer 1 Report
To the authors
There is a tremendous interest in the mechanisms associated with the very dramatic impact HEMT has had on the lives of pwCF. Thus, the title, and expected material to be presented in this review is very welcome. However, the topic is covered amidst a very lengthy discussion of many other areas of CF pathogenesis - not especially relevant to HEMT. The impact of HEMT and expected mechanism of action are provided - but not in an accessible fashion.
Despite the title, there is relatively relatively minimal discussion of the now abundant literature of outcomes of pwCF on therapy. This is- major weakness as HEMT has been revolutionary therapy. A more robust discussion of the impact on- numbers of transplants - lack of sputum production and changes in therapy is really needed for the reader to understand this therapy. The next question, which is the mechanism of action - then can be addressed in a more organized fashion.
Major questions are not directly addressed:
1- Does CFTR dysfunction cause an inherent immune defect that is further amplified by infection? This is central to understanding the effect of HEMT. Several relevant studies are mentioned - (Bonfield's bone marrow chimeras) but other important studies (Accurso's inflammation in infants study, or Puchelle's study of CF abortuses) are not. The Immunity article linking PTEN and CFTR as well as work from Bruscia's group and HO-1 are also relevant and could be better explained. As whether inflammation is a component of CFTR dysfunction is a critical issue - this information needs a separate section and better organization.
2- There is minimal discussion of HEMT use in pwCF with rare genotypes - this needs to be included and explained.
3- The discussion of HEMT and P. aeruginosa is complicated. The authors explain that there are major differences in early and late strains of bacteria isolated from CF patients. A major factor - that LPS expression and immunogenicity is modified over time need to be clarified although some of the relevant the references are mentioned; this is likely more relevant than flagella - which are repeatedly mentioned.
4- What about the host response? How does HEMT change this in vivo? There are exciting new data using scRNASeq to look at the host response to infection. These would be important to include (Britto AJCCM) - These human studies are more relevant than the many in vitro studies that are cited - using concentrated undefined materials from stored sputum. The in vitro data needs to be correlated with what has been observed in real patients.
5. l 396 - Are the neutrophils or monocytes actually "defective? New studies suggest this may be the case. However, amidst the lengthy discussion of in vitro - defective MPO activity - the authors need to point out that CF neutrophils are only defective in the lungs - not elsewhere - needs to be emphasized.
6. There is a large section on nutritional immunity - buried ( l9-10) in the discussion of neutrophil elastase. Ultimately - since the organisms readily cause persistent infection - they can acquire all of these nutrients . This section is interesting -but it is unclear how important this is to the focus of the review.
7. In writing review articles it is expected that the authors highlight their own work. However this much be put in context. The discussion of how HEMT alters mucociliary clearance and hence bacterial clearance (or not) needs to be better explained. For example,
l 560 there is a major bias that defective muco-ciliary clearance is the reason for infection - not entirely justified - the anti-bacterial effects provided are weak studies and not especially relevant
8. In section 6-3 the authors suggest that inflammation is "good" for CF therapy - really needs in vivo verification - SMM as a stimulus is not at all widely accepted (nor verified by other groups) - as it represents an undefined mixture of many immunostimulants. It is especially un likely to be relevant to the much more normal airway biology now commonplace in CF patients on modulators
Reviewer 2 Report
This is a well-written and comprehensive review of host-pathogen interactions in cystic fibrosis (CF). The review could benefit from editing specific subsections (see below). Furthermore, although it is not possible to cite every study, there are some key references/studies that should be included in this review.
Minor comments:
1) Subsection 4.3. is extremely long and detailed. This can be edited to improve clarity and readability.
2) Many of the important studies with ivacaftor in CF animal models (e.g., rat and ferret) are not included. The ferret CFTR-knockout model has also shed important insight into infection and inflammation in the early CF lung.
3) A recent report by Rehman et al., JCI, 2021 also supports enhanced CFTR rescue by inflammation (subsection 6.3). However, it should probably be mentioned that other inflammatory cytokines (e.g., TGF-beta1) can reduce the efficacy of modulators.
